# Hotspot analysis by confocal microscopy can help to differentiate challenging melanocytic skin lesions

Raquel de Paula Ramos Castro[1]*, Juliana Casagrande Tavoloni Braga[1], Mariana Petaccia de Macedo[2], Clóvis Antonio Lopes Pinto[2], José Humberto Tavares Guerreiro Fregnani[3], Gisele Gargantini Rezze[4]

1 Department of Cutaneous Oncology, AC Camargo Cancer Center, São Paulo, Brazil, 2 Department of Pathology, AC Camargo Cancer Center, São Paulo, Brazil, 3 Teaching and Research Institute, AC Camargo Cancer Center, São Paulo, Brazil, 4 Dermaimage Medical Associates, São Paulo, Brazil

* rprcastro@yahoo.com.br

**Data Availability Statement:** All relevant data are within the paper and its Supporting Information files.

## Abstract

Some melanocytic lesions do not present enough clinical and dermoscopic features to allow ruling out a possible melanoma diagnosis. These "doubtful melanocytic lesions" pose a very common and challenging scenario in clinical practice and were selected at this study for reflectance confocal microscopy evaluation and subsequent surgical excision for histopathological diagnosis. The study included 110 lesions and three confocal features were statistically able to distinguish benign melanocytic lesions from melanomas: "peripheral hotspot at dermo-epidermal junction", "nucleated roundish cells at the dermo-epidermal junction" and "sheet of cells". The finding of a peripheral hotspot (atypical cells in 1mm$^2$) at the DEJ is highlighted because has not been previously reported in the literature as a confocal feature related to melanomas.

## Introduction

The diagnosis of melanocytic skin lesions based only on clinical and dermoscopic evaluation can be challenging, even for experienced dermatologists. Some melanocytic lesions do not present enough clinical and dermoscopic features to establish a definitive diagnosis with reliability, imposing the need for biopsy and histopathological evaluation.

The use of additional non-invasive imaging techniques, like *in vivo* reflectance confocal microscopy (RCM) permits a cytoarchitectural evaluation of the epidermis, the dermo-epidermal junction (DEJ) and the upper dermis. Cellular atypia and pleomorphism can also be visualized *in vivo* to aid the diagnosis [1]. As a result, RCM represents a sensitive and specific tool for the early detection of melanomas and other skin tumors [1].

However, only few articles in the literature have described the RCM features present in melanocytic lesions. This study aimed to describe in detail the RCM features of melanocytic lesions with a doubtful diagnosis after clinical and dermoscopic evaluation, and search for new RCM features capable to differentiate them.

**Funding:** No received specific funding.

**Competing interests:** The authors have declared that no competing interests exist.

## Materials and methods

This retrospective study was approved by the Fundação Antônio Prudente ethical committee (01524/11) and all patients included agreed to participate and signed the informed consent document. A total of 96 patients from the Cutaneous Oncology Department of AC Camargo Cancer Center, São Paulo, Brazil, with 110 doubtful clinical and dermoscopic melanocytic lesions were selected. Only lesions suspicious of superficial spreading melanomas were included because other melanomas subtypes have other dermoscopy and confocal parameters (i.e. lentigo maligna, amelanotic melanoma, nodular melanomas and acral melanomas). Also, lesions located at sites where the Vivascope 1500 RCM device could not be adapted and lesions located at special sites such as the face, scalp and digits were excluded.

The dermoscopic diagnostic method used was the Pattern Analysis, applying the following criteria: eccentric pigmentation, abrupt network loss, poorly defined network, enlarged/atypical pigment network, multiple brown or dark globules with irregular shape and distribution, peppering, multiple colors, negative network, blue-white veil, radial streaks, pseudopods and structureless areas [2]. Melanocytic lesions with few or faint dermoscopy features related to melanoma diagnosis and lacked enough criteria for benign lesion were termed as "doubtful melanocytic lesions".

These "doubtful melanocytic lesions" were submitted to RCM examination by an experienced dermatologist using the VivaScope® 1500 confocal microscope (Lucid-Tech, Rochester, New York, USA). The examination was carried out in a step-by-step manner, with complete image documentation for subsequent analyses blinded from anatomopathological results. The RCM evaluation was based on features previously described (Table 1) [3–5].

In superficial spreading melanoma, junctional aggregates of melanocytes are commonly found, with high shape and size variability, composed of highly atypical melanocytes and with a tendency to confluence [6]. Due to the non-uniform distribution of these atypical melanocytes aggregates throughout the lesion and in order to quantify the degree and location of higher atypia inside a given lesion, hotspot analyses were included. A hotspot was defined as the 01 x 01 mm area of the lesion where the atypical cells are more aggregated, and was searched at the epidermis and DEJ levels. The selection of the hotspot was defined subjectively by the dermatologist after careful visual inspection of RCM mosaic images from the epidermis and DEJ levels, delimiting the 01 x 01 mm area with higher atypia and classifying its location as central or peripheral. The degree of atypia in a hotspot was based on the number of atypical cells inside its 01 x 01 mm area (absent, ≤ 10 or > 10 atypical cells). Illustrative cases of hotspot location and quantification are shown in Figs 1 and 2.

Subsequently, all lesions were excised for histopathological diagnosis, which was considered the gold standard for final diagnosis, and categorized into benign melanocytic lesions (common melanocytic nevi and atypical melanocytic nevi) or melanomas. Statistical analysis by simple and multiple logistic regressions were conducted using SPSS software (version 21) and p value ≤ 0.05 was considered significant.

## Results

All 110 melanocytic lesions were subjected to histopathological examination, and their final diagnoses were: 31 common nevi (28%), 53 atypical (dysplastic) nevi (48%) and 26 melanomas (24%). Intense atypia was present in 3 (6%) dysplastic nevi. Among the 26 melanomas, 18 (69%) were *in situ*, 7 (27%) were thin (Breslow < 1 mm) and 1 (4%) had Breslow of 1.42 mm. Seven (27%) melanomas appeared from a pre-existing nevus.

The features found in the RCM examination were analyzed according to the final histopathological diagnoses (Table 2).

**Table 1. Description of RCM features.**

| RCM features | Description |
|---|---|
| Honeycomb pattern at suprabasal epidermis: Typical/Atypical | Typical: normal or "preserved" honeycomb pattern; keratinocytes are well-demarcated, "visible". Atypical: "partial loss" of honeycomb pattern, keratinocytes demarcation are "poorly or not visible" |
| Atypical cells at epidermis: Presence/Absence | Atypical cells are large, bright and pleomorphic cells |
| Atypical cells at epidermis: Nucleated roundish/Dendritic | If present, atypical cells were classified: a) nucleated roundish cells—dark nucleus and bright cytoplasm, frequently twice the size of keratinocytes; b) dendritic cells—elongated branching structure extending from the cell body, usually present in melanocytes and Langerhans cells |
| Hotspot at epidermis: Presence/Absence | Hotspot was defined as the 1 x 1 mm area of the lesion that presents more atypia |
| Atypical cells at hotspot (epidermis): $\leq 10$ or $> 10$ | If present, hotspots were classified according to the number of atypical cells ($\leq 10$ or $> 10$ cells) |
| Location of hotspot at epidermis: Central/Peripheral | If present, hotspots were classified according to their predominantly location at the lesion: central or peripheral |
| Cobblestone pattern at basal cells: Typical/Atypical | Typical: uniform basal cells distribution; no variation in brightness or cellular outline between individual cells. Atypical: basal cells are not uniformly distributed |
| Papillae at DEJ: Edged/Non-edged | Edged: demarcated by a rim of bright basal cells (confluent cells). Non-edged: absence of a demarcated rim of bright cells, but separated by a series of large reflecting cells |
| General atypia at DEJ: Presence/Absence | Present if the normal architecture of the DEJ is partially or completely lost. Described as present if some RCM findings were noted: atypical meshwork pattern, atypical cells (dendritic or roundish cells), sheet of cells and "mitochondria-like structures" |
| Meshwork pattern at DEJ: Presence/Absence | Characterized by small dark holes surrounded by thickened interpapillary spaces |
| Meshwork pattern at DEJ: Typical/Atypical | If present, meshwork pattern was classified: a) typical—clearly thickened interpapillary spaces; b) atypical—irregular and enlarged interpapillary spaces by the presence of atypical cells |
| Atypical cells at DEJ: Presence/Absence | Atypical cells are large, bright and pleomorphic cells |
| Atypical cells at DEJ: Nucleated roundish/Dendritic | If present, atypical cells were classified: a) nucleated roundish cells—isolated round to oval refractive cells with a dark nucleus, located in the papillary dermis; b) dendritic cells—elongated dendritic cells around dermal papillae |
| Hotspot at DEJ: Presence/Absence | Hotspot was defined as the 1 x 1 mm area of the lesion that presents more atypia |
| Atypical cells at hotspot (DEJ): $\leq 10$ or $> 10$ | If present, hotspots were classified according to the number of atypical cells ($\leq 10$ or $> 10$ cells) |
| Location of hotspot at DEJ: Central/Peripheral | If present, hotspots were classified according to their predominantly location at the lesion: central or peripheral |
| Junctional nests: Presence/Absence | Oval compact cellular aggregates, bulging within the dermal papillae connected with the epidermal basal cell layer |
| Dense and sparse nests: Presence/Absence | Roundish nonreflecting structures with a well-demarcated border, containing isolated round to oval cells with dark nucleus and reflecting cytoplasm; sometimes presenting in a multilobate configuration |
| Dense (homogeneous) nests: Presence/Absence | Compact aggregates with sharp margin and similar cells in morphology and refractivity |
| Atypical nests: Presence/Absence | Dense and sparse nests composed by pleomorphic atypical cells |
| Peripheral nests: Presence/Absence | Enlarging nevus characterized by bulging junctional nests at the lesion periphery |
| Sheet of cells: Presence/Absence | Atypical pleomorphic melanocytes distributed in sheet-like structures |
| "Mitochondria-like structures": Presence/Absence | Elongated dendritic cells crowded around dermal papillae, some of them forming bridges that resembles the mitochondrial aspect |

(*Continued*)

**Table 1.** (Continued)

| RCM features | Description |
|---|---|
| Short interconnections: Presence/Absence | Junctional thickenings and nests surrounding the papillae |
| Inflammatory cells: Presence/Absence | Bright particles within the papillae |
| Melanophages: Presence/Absence | Plump irregularly shaped bright cells with ill-defined borders and usually no visible nucleus in single units or in clusters |
| Coarse collagen fibers: Presence/Absence | Collagen fibers are packed together forming a coarse web-like architecture |

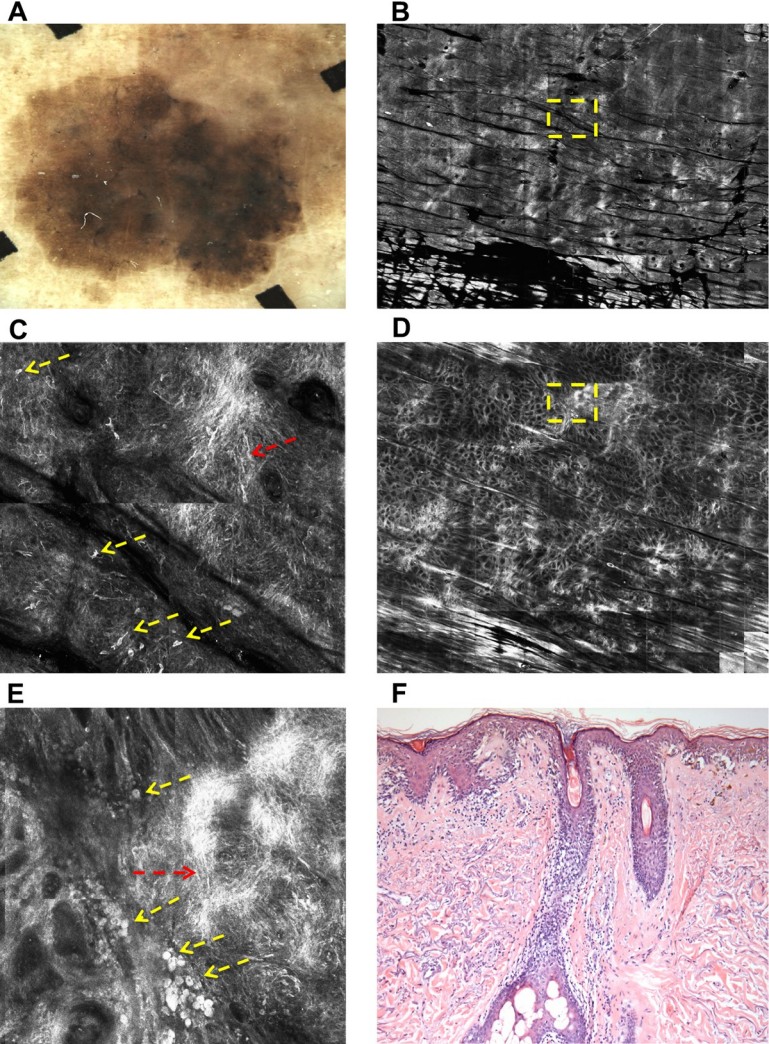

**Fig 1. Superficial spreading melanoma *in situ*. (A):** Dermoscopy shows atypical network and peppering. **(B):** RCM mosaic image (8.0 x 8.0 mm) at the spinous layer; visual inspection positioned the hotspot at a central location (1.0 x 1.0 mm square area is marked with yellow dashed outline and better shown in C). **(C):** Atypical honeycomb pattern and widespread roundish cells (yellow arrows) and dendritic cells (red arrows) in sheet of cells distribution. **(D):** RCM mosaic image (8.0 X 8.0 mm) at DEJ; visual inspection positioned the hotspot at a peripheral location (1.0 x 1.0 mm square area is marked with yellow dashed outline and better shown in E). **(E):** Non-edged papillae, roundish cells at DEJ (yellow arrows) and dendritic cells (red arrows) in sheet of cells distribution. **(F):** Histopathology confirms a superficial spreading melanoma *in situ* (H&E, original magnification x200), with disarrangement of the rete ridge and increased number of atypical melanocytes affecting the adnexae.

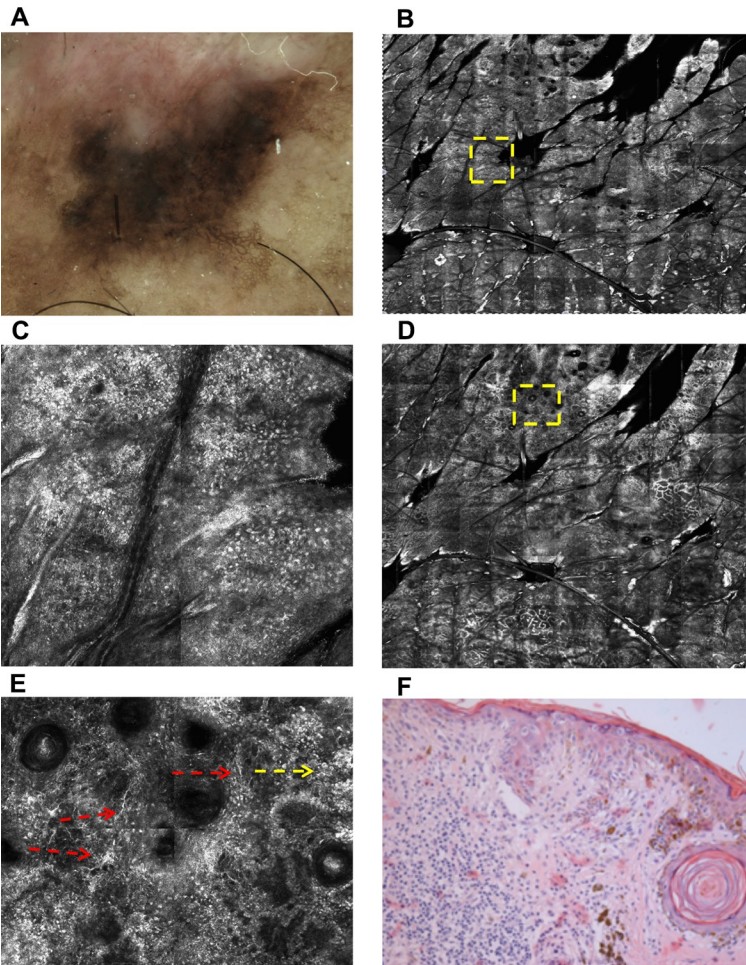

**Fig 2. Superficial spreading melanoma. (A):** Dermoscopy shows atypical network and peppering. **(B):** RCM mosaic image (8.0 x 8.0 mm) at epidermis (suprabasal layer); visual inspection positioned the hotspot at a central location (1.0 x 1.0 mm square area is marked with yellow dashed outline and better shown in C). **(C):** Atypical cobblestone pattern. **(D):** RCM mosaic image (8.0 X 8.0 mm) at DEJ; visual inspection positioned the hotspot at a peripheral location (1.0 x 1.0 mm square area is marked with yellow dashed outline and better shown in E). **(E):** Non-edged papillae, roundish cells at DEJ (yellow arrow) and dendritic cells (red arrows). **(F):** Histopathology confirms a superficial spreading melanoma (H&E, original magnification x200), with Breslow = 0.55 and presence of pagetoid cells.

Statistical analyses of RCM results were performed by simple and multiple logistic regressions. Several RCM findings at the DEJ occurred simultaneously and were labeled as "DEJ general atypia" to better differentiate the groups: atypical meshwork pattern, presence of atypical cells (dendritic or roundish cells), sheet of cells and "mitochondria-like structures".

After multiple logistic regression, the presence of three of these features remained statistically significant to differentiate benign melanocytic lesions (common and atypical melanocytic nevi) from melanomas: atypical roundish nucleated cells at DEJ (p = 0.048, 95% CI = 1.01–47.96), peripheral hotspot at DEJ (p = 0.032, 95% CI = 1.18–42.88) and sheet of cells (p = 0.04, 95% CI = 1.09–42.35) (Table 3). Only the presence of peripheral hotspot at DEJ was significantly related to melanoma diagnosis, instead predominantly central location of atypical cells at the DEJ hotspot was not.

Based on the final histological diagnosis, each different combination of these statistically significant RCM features was related to the probability of the lesion being a melanoma. The absence of these criteria confers a probability of 5.5%, while lesions that presented only one

**Table 2. Frequency of RCM features in common nevi, atypical nevi and melanomas.**

| RCM features | Common nevus (31) | Atypical nevus (53) | Melanoma (26) |
|---|---|---|---|
| Typical Honeycomb | 31 (100%) | 48 (91%) | 23 (43%) |
| Atypical Honeycomb | 0 (0%) | 5 (9%) | 3 (12%) |
| Atypical cells (nucleated roundish or dendritic cells) at epidermis | 07 (23%) | 32 (60%) | 23 (88%) |
| Nucleated roundish cells at epidermis | 0 (0%) | 2 (4%) | 4 (15%) |
| Dendritic cells at epidermis | 6 (19%) | 31 (58%) | 20 (77%) |
| Hotspot ≤ 10 atypical cells at epidermis | 6 (19%) | 33 (62%) | 9 (35%) |
| Hotspot > 10 atypical cells at epidermis | 0 (0%) | 15 (28%) | 14 (54%) |
| Absent of hotspot at epidermis | 25 (81%) | 5 (09%) | 3 (12%) |
| Central hotspot at epidermis | 5 (16%) | 44 (83%) | 16 (62%) |
| Peripheral at epidermis | 1 (3%) | 05 (6%) | 07 (27%) |
| Typical cobblestone | 31 (100%) | 42 (79%) | 12 (46%) |
| Atypical cobblestone | 0 (0%) | 11 (21%) | 14 (54%) |
| Edged papillae | 29 (94%) | 24 (45%) | 6 (23%) |
| Non-edged papillae | 2 (6%) | 29 (55%) | 20 (77%) |
| DEJ general atypia | 2 (6%) | 39 (74%) | 26 (100%) |
| Absent of meshwork pattern | 11 (35%) | 23 (43%) | 13 (50%) |
| Typical meshwork pattern | 20 (65%) | 17 (32%) | 4 (15%) |
| Atypical meshwork pattern | 0 (0%) | 13 (25%) | 9 (35%) |
| Atypical cells (nucleated roundish or dendritic cells) at DEJ | 2 (6%) | 29 (55%) | 20 (77%) |
| Nucleated roundish cells at DEJ | 0 (0%) | 6 (11%) | 2 (8%) |
| Dendritic cells at DEJ | 2 (6%) | 27 (51%) | 17 (65%) |
| Hotspot ≤ 10 atypical cells at DEJ | 2 (6%) | 17 (32%) | 3 (12%) |
| Hotspot > 10 atypical cells at DEJ | 0 (0%) | 12 (23%) | 15 (58%) |
| Absent of hotspot at DEJ | 29 (94%) | 24 (45%) | 8 (31%) |
| Central hotspot at DEJ | 2 (6%) | 20 (38%) | 12 (46%) |
| Peripheral at DEJ | 0 (0%) | 8 (15%) | 6 (23%) |
| Junctional nests | 23 (74%) | 47 (89%) | 23 (88%) |
| Dense and sparse nests | 11 (39%) | 15 (35%) | 9 (44%) |
| Dense (homogeneous) nests | 27 (30%) | 51 (40%) | 25 (60%) |
| Atypical nests | 0 (0%) | 3 (10%) | 6 (33%) |
| Peripheral nests | 10 (36%) | 17 (45%) | 5 (28%) |
| Sheet of cells | 0 (0%) | 2 (6%) | 7 (6%) |
| Mitochondria-like structures" | 0 (0%) | 6 (11%) | 9 (35%) |
| Short interconnections | 4 (13%) | 16 (30%) | 11 (42%) |
| Inflammatory cells | 14 (45%) | 37 (70%) | 22 (85%) |
| Melanophages | 13 (42%) | 26 (49%) | 16 (62%) |
| Melanophages isolated | 13 (42%) | 16 (30%) | 16 (62%) |
| Melanophages clusters | 4 (13%) | 15 (28%) | 09 (35%) |
| Coarse collagen fibers | 31 (100%) | 52 (98%) | 25 (96%) |

positive criterion presented a 31.5% to 37.5% probability, the lesions with two positive criteria presented 79.5% to 83.9% probability and the lesions that had three positive features presented 97.5% probability of being melanoma (Table 4).

## Discussion

The main advantage of RCM is increasing the sensitivity and specificity of melanoma diagnosis and differentiating them from common and atypical nevi [4, 7, 8]. "Featureless

**Table 3. Simple and multiple logistic regressions comparing RCM criteria in common and atypical nevi versus melanomas.**

**Simple logistic regression**

| RCM features | Category | OR | 95% CI | P value |
|---|---|---|---|---|
| Honeycomb | Typical or atypical pattern | 2.872 | 0.71–11.61 | 0.139 |
| Cobblestone | Typical pattern | 1.0 | 2.86–21.00 | <0.0001 |
| Cobblestone | Atypical pattern | 7.742 | 2.86–21.00 | <0.0001 |
| Atypical cells at epidermis | Nucleated roundish cells | 7.455 | 1.28–43.39 | 0.025 |
| Atypical cells at epidermis | Dendritic cells | 4.234 | 1.54–11.61 | 0.005 |
| Atypical cells at epidermis | Nucleated roundish cells or dendritic cells | 8.846 | 2.47–31.73 | 0.001 |
| Hotspot at epidermis | Absent | NA | NA | NA |
| Hotspot at epidermis | ≤ 10 cells | 5.423 | 1.35–21.81 | 0.017 |
| Hotspot at epidermis | > 10 cells | 19.939 | 4.87–81.61 | <0.0001 |
| Hotspot location at epidermis | Central | 7.667 | 2.06–2.05 | 0.002 |
| Hotspot location at epidermis | Peripheral | 17.889 | 3.62–88.41 | <0.0001 |
| Papillae at DEJ | Non-edged papillae | 5.699 | 2.07–15.71 | 0.001 |
| DEJ general atypia | Presence | NA | NA | NA |
| Meshwork | Absent | NA | NA | NA |
| Meshwork | Typical pattern | 0.283 | 0.08–0.95 | 0.041 |
| Meshwork | Atypical pattern | 1.811 | 0.63–5.24 | 0.27 |
| Atypical cells at DEJ | Nucleated roundish cells | 12.3 | 2.31–65.55 | 0.003 |
| Atypical cells at DEJ | Dendritic cells | 3.582 | 1.42–9.03 | 0.007 |
| Atypical cells at DEJ | Nucleated roundish cells or dendritic cells | 5.699 | 2.07–15.71 | 0.001 |
| Hotspot at DEJ | Absent | NA | NA | NA |
| Hotspot at DEJ | ≤10 cells | 1.046 | 0.25–4.36 | 0.951 |
| Hotspot at DEJ | > 10 cells | 8.281 | 2.86–23.96 | <0.0001 |
| Hotspot location at DEJ | Central | 3.682 | 1.32–10.24 | 0.012 |
| Hotspot location at DEJ | Peripheral | 5.062 | 1.39–18.45 | 0.014 |
| Junctional nest | Presence | 1.533 | 0.40–5.82 | 0.53 |
| Dense and homogeneous nests | Presence | 0.446 | 0.17–1.17 | 0.102 |
| Dense and sparse nests | Presence | 1.181 | 0.47–2.10 | 0.726 |
| Atypical nests | Presence | 8.1 | 1.86–35.22 | 0.005 |
| Location of nests | DEJ /dermis | 0.603 | 0.24–1.52 | 0.284 |
| Location of nests | Peripheral | 0.503 | 0.17–1.48 | 0.211 |
| Sheet of cells | Presence | 15.105 | 2.90–78.56 | 0.001 |
| "Mitochondria-like structures" | Presence | 6.882 | 2.16–21.92 | 0.001 |
| Short interconnections | Presence | 2.347 | 0.93–5.92 | 0.071 |
| Inflammatory cells | Presence | 3.559 | 1.13–11.26 | 0.031 |
| Melanophages | Presence | 1.846 | 0.75–4.54 | 0.181 |
| Melanophages | Isolated | 1.846 | 0.75–4.54 | 0.181 |
| Melanophages | Clusters | 1.811 | 0.70–4.71 | 0.223 |
| Coarse collagen fibers | Presence | 3.322 | 0.20–55.02 | 0.402 |

**Multiple logistic regression** [*]

| RCM features | Category | OR | 95% CI | P value |
|---|---|---|---|---|
| Atypical cells at DEJ | Nucleated roundish cells | 6.973 | 1.01–47.96 | 0.048 |
| Hotspot location at DEJ | Peripheral | 7.106 | 1.18–42.88 | 0.032 |
| Sheet of cells | Presence | 6.792 | 1.09–42.35 | 0.04 |

NA = not applicable

[*] shows only RCM features with p ≤ 0.05.

**Table 4. Probability of melanoma diagnosis, related to the presence of RCM features.**

| RCM features | Presence/Absence | | | | | | | |
|---|---|---|---|---|---|---|---|---|
| Sheet of cells | ✗ | ✓ | ✗ | ✗ | ✓ | ✓ | ✗ | ✓ |
| Nucleated roundish cells at DEJ | ✗ | ✗ | ✓ | ✗ | ✓ | ✗ | ✓ | ✓ |
| Peripheral hotspot at DEJ | ✗ | ✗ | ✗ | ✓ | ✗ | ✓ | ✓ | ✓ |
| **Melanoma diagnosis probability** | **5.5%** | **31.5%** | **33.5%** | **37.5%** | **79.5%** | **82.4%** | **83.9%** | **97.5%** |

✗ = absent, ✓ = present.

melanomas", a term used in literature to describe melanomas without clinical and dermoscopy signs of malignancy, and melanomas originating from pre-existing nevi, which may have only focal atypia [9], might also benefit from RCM evaluation [10].

With these advantages, RCM can be used with clinical examination and dermoscopy to decrease the number of unnecessary excisions. The ratio of surgically excised benign lesions to reach a single melanoma diagnosis has been related to examination method, with 45:1 by clinical examination [7–10], 17:1 by dermoscopy [10], 4–7:1 by dermoscopy associated to digital monitoring [7–10], and 2:1 by RCM [11].

Some algorithms including RCM evaluation have been proposed in the literature to reach melanoma diagnosis and were related to different sensitivity (86.1% - 93%) and specificity (57–95.3%) [8, 12, 13]. However, the majority of these studies included several subtypes of melanomas, as invasive and amelanotic melanomas.

In our study, we decided to include only melanocytic lesions with few or faint dermoscopy features related to melanoma diagnosis and lacked enough criteria for benign lesion, termed here as "doubtful melanocytic lesions", because they pose a very common and challenging scenario in clinical practice. Melanocytic lesions that had enough clinical and dermoscopy features to be classified with high probability of being benign or malignant were excluded because their management is already defined (follow-up or excision, respectively). As a result, our approach was highly effective to detect melanomas at their initial stage, with the majority of our melanomas identified as *in situ* (18 out of 26). After simple and multiple logistic regressions, three confocal features were statistically able to distinguish benign melanocytic lesions from melanomas.

The distribution of melanocytes forming aggregates is a frequently pattern found in melanocytic lesions. In benign melanocytic lesions, these aggregates are generally uniform spread throughout the lesion and constituted by non-atypical melanocytes [6]. In superficial spreading melanoma, the melanocyte aggregates are haphazardly distributed, with often highly variable shape and size, constituted by atypical melanocytes, tendency to confluence and located usually at the dermal epidermal junction and / or within the mid-portion and upper levels of the epidermis [14]. Our methods included the hotspot location to study the non-uniform distribution of these atypical melanocytes aggregates at the doubtful melanocytic lesions and also its degree of atypia.

The finding of a peripheral hotspot (atypical cells in 1mm$^2$) at the DEJ was revealed as a specific confocal feature for melanoma diagnosis, however with low sensitivity. On the other hand, the presence of a central hotspot was not statistically more frequent in melanomas. Central hotspots were seen in 12 melanomas (46%) and in 22 (26%) benign melanocytic lesions, while peripheral hotspots were noted in six melanomas (23%) and not in any benign melanocytic lesions. The description of this RCM feature has not been previously reported in the literature.

The detection of atypical nucleated roundish cells at the DEJ have already been reported by other studies and related to 15 times greater risk for a lesion being malignant. The presence of

cells distributed in sheet-like structures disrupting the papillary architecture of the basal layer was also reported as highly specific, but with low sensitivity for melanoma diagnosis [7]. These findings were confirmed in our study: atypical nucleated roundish cells at the DEJ was present in 6 (23.08%) melanomas and in 2 (2.38%) benign lesions (p = 0.048, 95% CI = 1.01–47.96); and "sheet of cells" was present in 7 (26.92%) melanomas and in 2 (2.38%) benign lesions (p = 0.04, 95% CI = 1.09–42.35).

## Conclusion

Our study is focused in RCM evaluation of doubtful melanocytic lesions and identified three confocal features statistically able to distinguish benign melanocytic lesions from melanomas. Among them, the finding of a peripheral hotspot (atypical cells in 1mm$^2$) at the DEJ is highlighted because has not been previously reported in the literature. Our study included a limited number of cases from a single institution and further research is still necessary to validate to importance and impact of this RCM feature to improve melanoma diagnosis accuracy.

## Supporting information

**S1 Table. Dermoscopy features of all lesions included at the study.**
(XLSX)

**S2 Table. Confocal microscopy features of all lesions included at the study.**
(XLSX)

## Author Contributions

**Formal analysis:** José Humberto Tavares Guerreiro Fregnani.

**Methodology:** Juliana Casagrande Tavoloni Braga, Mariana Petaccia de Macedo, Clóvis Antonio Lopes Pinto.

**Supervision:** Gisele Gargantini Rezze.

**Writing – original draft:** Raquel de Paula Ramos Castro.

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
