## [Decision Letter · Decision Letter 0]

21 Jul 2021

PONE-D-21-14949

A NEW REFLECTANCE CONFOCAL MICROSCOPY ALGORITHM TO DIFFERENTIATE CHALLENGING MELANOCYTIC SKIN LESIONS

PLOS ONE

Dear Dr. Castro,

Thank you for submitting your manuscript to PLOS ONE. After careful consideration, we feel that it has merit but does not fully meet PLOS ONE’s publication criteria as it currently stands. Therefore, we invite you to submit a revised version of the manuscript that addresses the points raised during the review process.

This manuscript is showing an interesting and relevant observation which may represent a novelty in the field: the identification of "hotspots" in RCM evaluation of melanocyte features for melanoma identification. In contrast, the development of another diagnostic algorithm does not appear to add much value to the field. In agreement with the expert reviewers, especially reviewer 1, this manuscript will benefit significantly from changing its focus to the definition and description of the new parameter ("hotspot") and its diagnostic value, positioned in the context of literature.

We look forward to receiving your revised manuscript.

Kind regards,

Nikolas K. Haass, MD/PhD

Academic Editor

PLOS ONE

Journal Requirements:

"The study was approved by the institution’s ethical committee (01524/11).

Patients included agreed to participate and signed the informed consent document. A total of 96 patients from the Cutaneous Oncology Department of AC Camargo Cancer Center, São Paulo, Brazil. The form of the consent was written.".   

Once you have amended this statement in the Methods section of the manuscript, please add the same text to the “Ethics Statement” field of the submission form (via “Edit Submission”).

Additional Editor Comments (if provided):

This manuscript is showing an interesting and relevant observation which may represent a novelty in the field: the identification of "hotspots" in RCM evaluation of melanocyte features for melanoma identification. In contrast, the development of another diagnostic algorithm does not appear to add much value to the field. In agreement with the expert reviewers, especially reviewer 1, this manuscript will benefit significantly from changing its focus to the definition and description of the new parameter ("hotspot") and its diagnostic value, positioned in the context of literature.

Reviewers' comments:

Reviewer's Responses to Questions

**Comments to the Author**

1. Is the manuscript technically sound, and do the data support the conclusions?

Reviewer #1: No

Reviewer #2: Partly

2. Has the statistical analysis been performed appropriately and rigorously? 

Reviewer #1: Yes

Reviewer #2: Yes

3. Have the authors made all data underlying the findings in their manuscript fully available?

Reviewer #1: Yes

Reviewer #2: Yes

4. Is the manuscript presented in an intelligible fashion and written in standard English?

Reviewer #1: No

Reviewer #2: Yes

5. Review Comments to the Author

Reviewer #1: Dear Authors

the paper proposes a new algorithm for melanoma diagnosis. There are some limitations:

Number of cases is little compared to recent literature. In a list of so many parameters selection of some and their combination is affected by a relevant statistical instability.

The "hotspot" parameter is newly introduced and not clearly described

Most of the parameters are likely to be each-other correlated (for example, if hotspot is an area where atypical cells are more densely packed, it should be assumed that there is correspondence with presence of atypical cells).

Lack of distinction of subgroups of lesions (palpable/nodular vs. fact; facial vs. body ..) where different patterns are typically used for differential diagnosis

I feel that the value of this paper is more in the identification of a new descriptor than in the introduction of another algorithm.

Thus my recommendation is to reconsider extensively the value of the paper, concentrating on the definition and description of the new parameter ("hotspot"), making easy and clear its identification, considering situation where there are numerous cells everywhere (would you call this hotspot or not? any situation with diffused atypical cells is also presenting an hotspot?), considering relative values (a melanoma LM type is usually presenting dendritic clells infiltrating the follicles: could be this an hotspot?)

Then, compare the value of "hotspot" in diagnosis of melanoma, considering this feature per se, comparing with previously proposed algorithms, and as an addition to other already used algorithms. Only if it could make sense, create your own algorithm.

Reviewer #2: Authors claim that their algorithm is better than the previous ones for melanoma. However, there is no proof to that. I would suggest the authors to perform diagnostic accuracy by their expert readers on the same used in the study using Pellacani’ s and other algorithm.

6. PLOS authors have the option to publish the peer review history of their article (what does this mean?). If published, this will include your full peer review and any attached files.

Reviewer #1: No

Reviewer #2: No

---

## [Author Response · Author response to Decision Letter 0]

1 Sep 2021

Dear editor and reviewers,

 We thank for the careful revision of our article and all comments. We accepted all suggestions and made changes accordingly.

AUTHORS: The manuscript was revised and modifications were made to meet the style requirements.

"The study was approved by the institution’s ethical committee (01524/11).

Patients included agreed to participate and signed the informed consent document. A total of 96 patients from the Cutaneous Oncology Department of AC Camargo Cancer Center, São Paulo, Brazil. The form of the consent was written.". 

Once you have amended this statement in the Methods section of the manuscript, please add the same text to the “Ethics Statement” field of the submission form (via “Edit Submission”).

AUTHORS: The full name of the ethics committee/institutional review board(s) that approved your specific study was included at the Materials and Methods section of the manuscript: Line 104: “This retrospective study was approved by the Fundação Antônio Prudente ethical committee (01524/11) and all patients included agreed to participate and signed the informed consent document.”

AUTHORS: The minimal data set underlying the results described in our manuscript were attached as supplemental files. Table 5 has the dermoscopy features of all lesions included at the study and Table 6 has the RCM features of all lesions included at the study.

AUTHORS: An ORCID iD for the corresponding author was created, and this information was included at the Title Page.

Additional Editor Comments (if provided):

This manuscript is showing an interesting and relevant observation which may represent a novelty in the field: the identification of "hotspots" in RCM evaluation of melanocyte features for melanoma identification. In contrast, the development of another diagnostic algorithm does not appear to add much value to the field. In agreement with the expert reviewers, especially reviewer 1, this manuscript will benefit significantly from changing its focus to the definition and description of the new parameter ("hotspot") and its diagnostic value, positioned in the context of literature.

AUTHORS: We agree with the comment, and the manuscript was modified to highlight the identification and importance of "peripheral hotspots at DEJ" in RCM evaluation for melanoma identification. Also, the development of another diagnostic algorithm was excluded from the manuscript. We believe that these modifications made, the manuscript is more concise, address more directly and is focused in your main contribution for the literature.

Comments to the Author

1. Is the manuscript technically sound, and do the data support the conclusions?

Reviewer #1: No

Reviewer #2: Partly

AUTHORS: Modifications were made in the manuscript to facilitate replication studies. At the methods section, the “hotspot feature” is better explained and figures were included to depict our finding. Also, all results form the 110 melanocytic lesions were included as supplemental files.

Lines 117-118: “Melanocytic lesions with few or faint dermoscopy features related to melanoma diagnosis and lacked enough criteria for benign lesion were termed as “doubtful melanocytic lesions”. 

Lines 126-136: “In superficial spreading melanoma, junctional aggregates of melanocytes are commonly found, with high shape and size variability, composed of highly atypical melanocytes and with a tendency to confluence (6). Due to the non-uniform distribution of these atypical melanocytes aggregates throughout the lesion and in order to quantify the degree and location of higher atypia inside a given lesion, hotspot analyses were included. A hotspot was defined as the 01 x 01 mm area of the lesion where the atypical cells are more aggregated, and was searched at the epidermis and DEJ levels. The selection of the hotspot was defined subjectively by the dermatologist after careful visual inspection of RCM mosaic images from the epidermis and DEJ levels, delimiting the 01 x 01 mm area with higher atypia and classifying its location as central or peripheral. The degree of atypia in a hotspot was based on the number of atypical cells inside its 01 x 01 mm area (absent, ≤ 10 or > 10 atypical cells). Illustrative cases of hotspot location and quantification are shown in figures 1 and 2”.

2. Has the statistical analysis been performed appropriately and rigorously? 

Reviewer #1: Yes

Reviewer #2: Yes

• AUTHORS: Our statistical analyses were performed by an statistician, and all results were also included as supplemental files (Tables 5 and 6).

3. Have the authors made all data underlying the findings in their manuscript fully available?

Reviewer #1: Yes

Reviewer #2: Yes

AUTHORS: In order to make all data underlying our findings fully available, supplemental files were included:

Table 5: Dermoscopy features of all lesions included at the study, 

Table 6: Confocal microscopy features of all lesions included at the study.

4. Is the manuscript presented in an intelligible fashion and written in standard English?

Reviewer #1: No 

Reviewer #2: Yes

AUTHORS: The manuscript text was revised and typographical or grammatical errors were corrected. 

5. Review Comments to the Author

Reviewer #1: Dear Authors

The paper proposes a new algorithm for melanoma diagnosis. There are some limitations:

Number of cases is little compared to recent literature. In a list of so many parameters selection of some and their combination is affected by a relevant statistical instability.

The "hotspot" parameter is newly introduced and not clearly described.

Most of the parameters are likely to be each-other correlated (for example, if hotspot is an area where atypical cells are more densely packed, it should be assumed that there is correspondence with presence of atypical cells).

Lack of distinction of subgroups of lesions (palpable/nodular vs. fact; facial vs. body ..) where different patterns are typically used for differential diagnosis-

I feel that the value of this paper is more in the identification of a new descriptor than in the introduction of another algorithm.

Thus my recommendation is to reconsider extensively the value of the paper, concentrating on the definition and description of the new parameter ("hotspot"), making easy and clear its identification, considering situation where there are numerous cells everywhere (would you call this hotspot or not? any situation with diffused atypical cells is also presenting an hotspot?), considering relative values (a melanoma LM type is usually presenting dendritic clells infiltrating the follicles: could be this an hotspot?)

Then, compare the value of "hotspot" in diagnosis of melanoma, considering this feature per se, comparing with previously proposed algorithms, and as an addition to other already used algorithms. Only if it could make sense, create your own algorithm.

AUTHORS: We thank you for the careful revision and comments. 

-We agree the number of lesions included is limited and from a single institution. We decided to remove from the manuscript the development of another diagnostic algorithm and focused on the description of the RCM features found at the lesions. This limitation is highlighted at the conclusions section:

Lines 250-252: “Our study included a limited number of cases from a single institution and further research is still necessary to validate to importance and impact of this RCM feature to improve melanoma diagnosis accuracy. “

-At the methods section, the “hotspot feature” is better explained and figures were included to depict this finding.

Lines 126-136: “In superficial spreading melanoma, junctional aggregates of melanocytes are commonly found, with high shape and size variability, composed of highly atypical melanocytes and with a tendency to confluence (6). Due to the non-uniform distribution of these atypical melanocytes aggregates throughout the lesion and in order to quantify the degree and location of higher atypia inside a given lesion, hotspot analyses were included. A hotspot was defined as the 01 x 01 mm area of the lesion where the atypical cells are more aggregated, and was searched at the epidermis and DEJ levels. The selection of the hotspot was defined subjectively by the dermatologist after careful visual inspection of RCM mosaic images from the epidermis and DEJ levels, delimiting the 01 x 01 mm area with higher atypia and classifying its location as central or peripheral. The degree of atypia in a hotspot was based on the number of atypical cells inside its 01 x 01 mm area (absent, ≤ 10 or > 10 atypical cells). Illustrative cases of hotspot location and quantification are shown in figures 1 and 2.”

-All lesions were analyzed for several RCM features, and some parameters are each-other correlated. However, only three RCM features were statistically able to distinguish benign melanocytic lesions from melanomas and the manuscript focused on their description and importance. If readers are also interested in more detail of all RCM features analyzed, we included all data as supplemental files.

-Different subtypes of melanomas were excluded because they have other dermoscopy and confocal parameters. The inclusion and exclusion criteria were better explained in the methods section:

Lines 107-112: “Only lesions suspicious of superficial spreading melanomas were included because other melanomas subtypes have other dermoscopy and confocal parameters (i.e. lentigo maligna, amelanotic melanoma, nodular melanomas and acral melanomas). Also, lesions located at sites where the Vivascope 1500 RCM device could not be adapted and lesions located at special sites such as the face, scalp and digits were excluded.”

-We agree with the recommendations and made modifications accordingly. This made the manuscript more concise and addressed more directly its main contribution for the literature (description of the “peripheral hotspot at the DEJ” feature). This feature is also better explained, to clarify where the hotspot should be located (“where the atypical cells are more aggregated”):

Lines 227-234: “The distribution of melanocytes forming aggregates is a frequently pattern found in melanocytic lesions. In benign melanocytic lesions, these aggregates are generally uniform spread throughout the lesion and constituted by non-atypical melanocytes (6). In superficial spreading melanoma, the melanocyte aggregates are haphazardly distributed, with often highly variable shape and size, constituted by atypical melanocytes, tendency to confluence and located usually at the dermal epidermal junction and / or within the mid-portion and upper levels of the epidermis (14). Our methods included the hotspot location to study the non-uniform distribution of these atypical melanocytes aggregates at the doubtful melanocytic lesions and also its degree of atypia.”

-At discussion, our findings are compared to other studies and algorithms of the literature. We decided to exclude the algorithm due to the limited number of cases.

Reviewer #2: Authors claim that their algorithm is better than the previous ones for melanoma. However, there is no proof to that. I would suggest the authors to perform diagnostic accuracy by their expert readers on the same used in the study using Pellacani’ s and other algorithm.

AUTHORS: We thank the reviewer for the comment and decided to exclude the algorithm due to the limited number of cases. Also, at discussion, our findings are compared to other studies and algorithms of the literature. This made the manuscript more concise and addressed more directly its main contribution for the literature (description of the “peripheral hotspot at the DEJ” feature). This feature was also better detailed:

Lines 126-136: “In superficial spreading melanoma, junctional aggregates of melanocytes are commonly found, with high shape and size variability, composed of highly atypical melanocytes and with a tendency to confluence (6). Due to the non-uniform distribution of these atypical melanocytes aggregates throughout the lesion and in order to quantify the degree and location of higher atypia inside a given lesion, hotspot analyses were included. A hotspot was defined as the 01 x 01 mm area of the lesion where the atypical cells are more aggregated, and was searched at the epidermis and DEJ levels. The selection of the hotspot was defined subjectively by the dermatologist after careful visual inspection of RCM mosaic images from the epidermis and DEJ levels, delimiting the 01 x 01 mm area with higher atypia and classifying its location as central or peripheral. The degree of atypia in a hotspot was based on the number of atypical cells inside its 01 x 01 mm area (absent, ≤ 10 or > 10 atypical cells). Illustrative cases of hotspot location and quantification are shown in figures 1 and 2.”

---

## [Decision Letter · Decision Letter 1]

28 Jan 2022

Hotspot analysis by confocal microscopy can help to differentiate challenging melanocytic skin lesions

PONE-D-21-14949R1

Dear Dr. Castro,

We’re pleased to inform you that your manuscript has been judged scientifically suitable for publication and will be formally accepted for publication once it meets all outstanding technical requirements.

Kind regards,

Nikolas K. Haass, MD/PhD

Academic Editor

PLOS ONE

Additional Editor Comments (optional):

The authors have addressed all comments.

Reviewers' comments:

Reviewer's Responses to Questions

**Comments to the Author**

1. If the authors have adequately addressed your comments raised in a previous round of review and you feel that this manuscript is now acceptable for publication, you may indicate that here to bypass the “Comments to the Author” section, enter your conflict of interest statement in the “Confidential to Editor” section, and submit your "Accept" recommendation.

Reviewer #1: All comments have been addressed

2. Is the manuscript technically sound, and do the data support the conclusions?

Reviewer #1: Yes

3. Has the statistical analysis been performed appropriately and rigorously? 

Reviewer #1: Yes

4. Have the authors made all data underlying the findings in their manuscript fully available?

Reviewer #1: Yes

5. Is the manuscript presented in an intelligible fashion and written in standard English?

Reviewer #1: Yes

6. Review Comments to the Author

Reviewer #1: All requests have been adequately addressed. No further comments or anything to add.

7. PLOS authors have the option to publish the peer review history of their article (what does this mean?). If published, this will include your full peer review and any attached files.

Reviewer #1: No

---

## [Editor Report · Acceptance letter]

4 Feb 2022

PONE-D-21-14949R1 

Hotspot analysis by confocal microscopy can help to differentiate challenging melanocytic skin lesions 

Dear Dr. Castro:

I'm pleased to inform you that your manuscript has been deemed suitable for publication in PLOS ONE. Congratulations! Your manuscript is now with our production department. 

Kind regards, 

on behalf of

Prof Nikolas K. Haass 

Academic Editor

PLOS ONE